# Dating ancient manuscripts using radiocarbon and AI-based writing style analysis

Mladen Popović[1]*, Maruf A. Dhali[1,2], Lambert Schomaker[2], Johannes van der Plicht[3], Kaare Lund Rasmussen[4], Jacopo La Nasa[5], Ilaria Degano[5], Maria Perla Colombini[5], Eibert Tigchelaar[6]

1 Qumran Institute, University of Groningen, Groningen, The Netherlands, 2 Artificial Intelligence, Bernoulli Institute, University of Groningen, Groningen, The Netherlands, 3 Center for Isotope Research, University of Groningen, Groningen, The Netherlands, 4 Department of Physics, Chemistry, and Pharmacy, University of Southern Denmark, Sønderborg, Denmark, 5 Department of Chemistry and Industrial Chemistry, University of Pisa, Pisa, PL, Italy, 6 Faculty of Theology and Religious Studies, KU Leuven, Leuven, Belgium

* m.popovic@rug.nl

**Data availability statement:** All data, code, and test film associated with this article are publicly

## Abstract

Determining by means of palaeography the chronology of ancient handwritten manuscripts such as the Dead Sea Scrolls is essential for reconstructing the evolution of ideas, but there is an almost complete lack of date-bearing manuscripts. To overcome this problem, we present Enoch, an AI-based date-prediction model, trained on the basis of 24 $^{14}$C-dated scroll samples. By applying Bayesian ridge regression on angular and allographic writing style feature vectors, Enoch could predict $^{14}$C-based dates with varied mean absolute errors (MAEs) of 27.9 to 30.7 years. In order to explore the viability of the character-shape based dating approach, the trained Enoch model then computed date predictions for 135 non-dated scrolls, aligning with 79% in palaeographic post-hoc evaluation. The $^{14}$C ranges and Enoch's style-based predictions are often older than traditionally assumed palaeographic estimates, leading to a new chronology of the scrolls and the re-dating of ancient Jewish key texts that contribute to current debates on Jewish and Christian origins.

## 1 Introduction

The discovery of the Dead Sea Scrolls from ancient Judaea fundamentally transformed our knowledge of Jewish and Christian origins [1]. Determining the chronology of these handwritten manuscripts, mostly written in Aramaic/Hebrew script, is essential for reconstructing the evolution of ideas. Palaeography—the study of ancient handwriting—is used to date manuscripts on the basis of their handwriting [2–4] but its subjectivity and the impossibility of measuring the significance of handwriting features poses a problem [5,6]. The model that was constructed to trace the evolution from the imperial Aramaic script (fifth and fourth centuries BCE) to the Jewish square script (first and second centuries CE) and to date scrolls and religious, cultural, and historical developments [7–10] is not reliably grounded. For

available on Zenodo with the following DOIs: - Data and prediction plots (v3): https://doi.org/10.5281/zenodo.10998958. - Code and feature files (v6): https://doi.org/10.5281/zenodo.13319794. - Film (see details in S7 Appendix): https://doi.org/10.5281/zenodo.8167946.

**Funding:** This project has received funding by the European Research Council under the European Union's Horizon 2020 research and innovation programme under grant agreement no. 640497 (HandsandBible). The funding was received by M. Popović. The funders had no role in study design, data collection and analysis, decision to publish, or preparation of the manuscript.

**Competing interests:** The authors have declared that no competing interests exist.

palaeographic comparison, one requires enough date-bearing manuscripts in similar script, but date-bearing documents are hardly available among the scrolls and most scrolls have no stratigraphy in the archaeological record. Only some of the very oldest and the very youngest manuscripts have calendar dates. Inscriptions or historical hypotheses, such as a slow development of the Aramaic/Hebrew script in the third century BCE and the emergence and rapid development of a national script around the mid-second century BCE [8,9,11,12], cannot compensate for the palaeographic scarcity for the centuries in between and do not enable one to reliably date the scrolls (see S1 Appendix).

In this study, we present new $^{14}$C dates derived from manuscript samples as reliable time markers to bridge the chronological gap between the fourth century BCE and the second century CE. The $^{14}$C dating leads to a list of expected dates for a number of chosen manuscript fragments, though this list cannot be ordered sequentially, independent of $^{14}$C. Based on the dating results for the tested samples, Enoch, named after the ancient Jewish science hero, was trained as a machine-learning-based date-prediction model applying Bayesian ridge regression on established handwriting-style descriptors. The corresponding handwritten style features in those tested manuscript images can then be used to estimate the date of undated manuscripts. Thus, Enoch reduces palaeographic subjectivity and the role of implicit knowledge by offering date predictions as probability-based options, grounded in physical ($^{14}$C) and geometric (shape-based) evidences, that can aid palaeographers and historians in their decision-making and contribute to historical debates.

## 2 Integration of multiple dating methods

### 2.1 Radiocarbon dating

We performed $^{14}$C dating on 30 manuscripts from 4 sites, spanning an estimated 5 centuries: 25 from the Qumran caves, 1 from Masada, 2 from the Murabba'at at caves, and 2 from the Naḥal Ḥever caves (see Sect 2.1 in S2 Appendix). This study is the first to apply to the scrolls, prior to $^{14}$C dating, a chemical treatment specifically designed to remove fatty materials, employing solvent extraction (see Sects 2.2 and 2.7.1 in S2 Appendix). Additional specialized analytical chemistry methods were applied before and after sample pretreatment to demonstrate that the total amount of lipid materials is below a threshold that does not significantly skew the $^{14}$C date (see Sects 2.7.2–2.7.5 in S2 Appendix). The samples were dated by two Accelerator Mass Spectrometry (AMS) machines (see Sect 2.3 in S2 Appendix). For the relevant time range, the calibrated results are often bimodal, which is an effect of the calibration curve not being monotonous, but this issue can be solved (see Sect 2.4 in S2 Appendix and Sects 5.6–5.7 in S5 Appendix).

The AMS results yielded 27 valid $^{14}$C dates (see Sects 2.4–2.6 in S2 Appendix and Tables in S1 Appendix), improving and extending the existing series of $^{14}$C-dated Dead Sea Scrolls [13, 14]. In general, the $^{14}$C results indicate older date ranges for individual manuscripts as well as for the emergence of the so-called 'Hasmonaean' and 'Herodian' scripts (see S4 Appendix). Fig 1 shows the comparison between the $2\sigma$ accepted calibrated ranges and traditional palaeographic estimates (in blue and red; please note that the accepted range is not the complete calibrated range; for more details, see Sect 4.2 in S4 Appendix and Sect 5.6 in S5 Appendix). Only two manuscripts have date ranges that go in the direction of a younger possible range. The $^{14}$C results for most manuscripts confirm the basic distinction between older Hasmonaean-type manuscripts and younger Herodian-style manuscripts, and also between so-called 'Archaic' and Hasmonaean-type manuscripts. However, the $^{14}$C date ranges for manuscripts traditionally considered Hasmonaean and Herodian are quite differently distributed throughout the timeline. As can be seen in Fig 1 (in blue), Hasmonaean-type

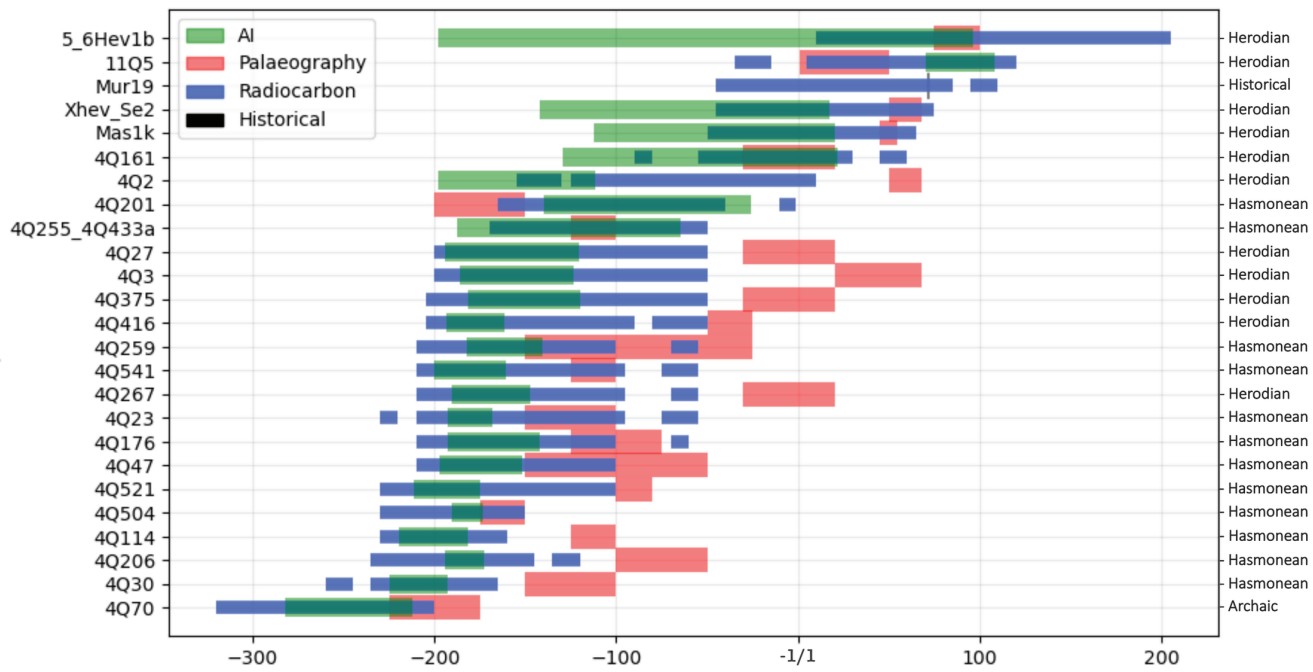

**Fig 1. Overview of date estimations by three information sources and a calendar date.** Blue bars indicate (accepted) $2\sigma$ calibrated ranges $^{14}$C, green indicates Enoch, red refers to palaeography, and black denotes the historical date. The vertical axis contains the manuscript numbers, and the horizontal axis contains dates: BCE in negative and CE in positive. Broad palaeographic types of the samples are indicated on the right.

manuscripts are all grouped together in a narrower part of the timeline but Herodian-type manuscripts are more spread across the timeline, extending from the second century CE all the way back to the second century BCE (see Sects 4.1.1–4.1.3 in S4 Appendix). The validated $^{14}$C results represent a data set that will be used in the next stage.

Sample 4Q114 is one of the most significant findings of the $^{14}$C results. The manuscript preserves Daniel 8–11, which scholars date on literary-historical grounds to the 160s BCE [15,16]. The accepted $2\sigma$ calibrated range for 4Q114, 230–160 BCE, overlaps with the period in which the final part of the biblical book of Daniel was presumably authored (see Sect 4.1.2 in S4 Appendix).

## 2.2 Date-prediction model

The completed $^{14}$C results set the stage for further manuscript dating analysis. The granularity of the $^{14}$C dating is coarse due to the limited number of data points but character-shape-based analysis provides an additional information source to tap into the historical development of handwriting styles.

While it is tempting to use modern methods of deep learning, as we have done before [17–21], there are several arguments to not currently use such approaches for the proposed style-based date prediction on a very small data set. It was decided to let the available data speak, not depending on a pre-trained model (S6 Appendix). Since a general, large, representative, and labeled data set is not available for the period of the scrolls, we apply dedicated pattern recognition and machine-learning models, only using the relevant scrolls data for training a date-prediction model. Given the topic's importance, pre-trained deep transfer learning based on extraneous material may be expected to elicit valid concerns among

palaeographers about the relation between the scrolls' target data and training data from a (very) different origin and period.

We used 24 manuscripts from the $^{14}$C samples with valid dates as a labeled dataset for the primary training set for Enoch (see Sects 2.4–2.6 in S2 Appendix and Sect 4.2 in S4 Appendix). For the data labels, we used OxCal v4.4.2 [22,23] to obtain the raw data points for the probability distributions. This is because the $^{14}$C results are not single dates, as with date-bearing documents, but represent date ranges with probability distributions. The $^{14}$C data input for training Enoch consists of the probability distributions of accepted $2\sigma$ calibrated ranges. In the case of bimodal evidence, we used a Bayesian method outside the OxCal program to limit the date ranges produced from each calibration. We did this to prevent propagation of $^{14}$C bimodality, which is an effect of the calibration curve not being monotonous (see Sect 2.4 in S2 Appendix), to the next processing stage of training the style-based predictor. Only in the case of bimodal evidence was palaeographic domain knowledge used in training the Enoch model as it allows making a binary split in the OxCal distribution of $2\sigma$ ranges using the Heaviside function with the position of the step being placed at an innocuous low-probability point, near zero, on the curve (see Sect 5.6 in S4 Appendix).

In addition to the primary training set, we created different combinations of training data to perform comparative analyses and further check the robustness of the model. These combinations include the tentative addition or omission of 4Q52, some previously tested $^{14}$C samples [13,14], date-bearing documents from the fifth–fourth centuries BCE and the second century CE (see tables in S9 Appendix for complete lists of manuscripts), the Maresha ostracon from 176 BCE (see Sect 1.1 in S1 Appendix). We performed cross-validation in three ways: using a train-validation split, via manuscript-image splitting for train-test sets, and through leave-one-out of the training data points.

**2.2.1 Deep neural networks for detection of handwritten ink-trace patterns** For this study, the digital images of the 24 $^{14}$C-dated manuscripts [24,25] underwent multiple pre-processing measures to become suitable for pattern recognition-based techniques. It should be noted that the images are extremely difficult to work with (some examples can be seen in illustration 9 in S5 Appendix and illustration 22 in S7 Appendix; see also Sect 7.1 in S7 Appendix). We are not dealing with digitally encoded text but with pixel images of highly degraded manuscripts as input.

We utilize multispectral band images of each fragment and employ BiNet [26], an artificial neural network based on an encoder-decoder U-net architecture designed to binarize the diverse range of scroll images, to generate a three-channel image. The resulting binarized images consist solely of black foreground pixels (ink) against a white background, ensuring that subsequent style analyses focus exclusively on the handwritten patterns while minimizing inadvertent matches due to material-texture attributes. We further correct the rotation of the binarized images and divide them into multiple parts to maintain a balanced distribution of handwritten characters within each new image. No extraneous image material was used to train this binarization method.

Thus, we obtained a data set of 75 images from the 24 $^{14}$C-dated manuscripts. The image samples typically contain 150–200 characters, which has been shown to be sufficient for the comparable task of writer identification [27].

**2.2.2 Extracting features for style attribution** In this study, 'style' is not related to textual content or wording. For characterizing the handwritten shapes, small shapes along the ink trace are used, largely uncoupled from the textual content, because we want to avoid spurious matches or date predictions on the basis of textual content. Once the training images were available, we could perform feature extraction techniques to translate handwriting patterns into feature vectors. The feature vectors relate directly to the shape-based evidence of

the ink traces in the manuscripts and have a solid basis in writer identification [28–30] and document dating [31,32]. We extract features from both the allographic and textural levels of characters [6]. An overview of machine-learning methods can be found in [33].

The allographic method uses a self-organized character map obtained using a Kohonen neural network. As an example, this allographic codebook feature allows for a 93% ($\pm \sigma$ = 2.3%) accuracy classification of the scripts 'Hasmonaean' vs. 'Herodian', using PCA, on 590 labeled manuscripts, results averaged over 32 random odd/even splits for training/testing [34]. The textural method uses statistical pattern recognition on angular information. The 'hinge' method for estimating the curvature distribution has been used extensively in writer verification and dating studies [30,31,35]. Whereas the allographic feature addresses stylistic elements at the character level, the 'hinge' method concerns a micro-level feature directly related to the original writing activity that yielded the curvature of the ink trace. Therefore, we make a weighted combination of textural and allographic features to obtain an adjoined feature vector for each manuscript image. Such a feature vector constitutes the input data to Enoch.

**2.2.3 Bayesian ridge regression**  Due to the limited size of the data set, we cannot employ high-parametric models like period-specific temporal codebooks [32]. Instead, we utilize conditional modeling with Bayesian ridge regression [36] (see Sect 5.6 in S5 Appendix). This approach applies Bayesian inference to estimate model parameters for date prediction. By placing a prior distribution on the parameters and updating it with observed data using Bayes' rule, we obtain the posterior distribution of the parameters and predicted dates. The Bayesian approach is chosen because our target output data represents probability curves for $^{14}$C dates (i.e., a vector) containing the accepted $2\sigma$ calibrated 'OxCal' ranges. This probabilistic approach enables us to incorporate all available information while maintaining interpretability. Moreover, instead of producing a single number for the estimated date of a sample, it provides a posterior distribution that allows us to assess the uncertainty associated with the estimated dates. Additionally, Enoch is able to provide error margins for predictions on unseen data. Regarding data balance, we used two methods to compensate for the imbalanced distribution of the training data over different periods (see Sect 5.7 in S5 Appendix).

# 3 Validating Enoch

We used 62 images of the data set of 75 images from the 24 $^{14}$C-dated manuscripts to train Enoch. We then validated Enoch in two ways. The remaining 13 images were passed as unseen test data to cross-validate the robustness and reliability of Enoch's performance. The prediction of these 13 images by Enoch gives an 85.14% overlap to the original $^{14}$C probability distributions (see tables in S9 Appendix). We also performed validation by leave-one-out tests (see Sect 5.9.1 in S5 Appendix). Fig 1 (in green) also shows the results of cross-validation and leave-one-out tests for training Enoch. The choice for the bandwidths ($2\sigma$ date ranges for $^{14}$C, $1\sigma$ uncertainties of the ridge regression for style-based predictions) is based on the intrinsic reliability of the two information sources. $^{14}$C date ranges are evidently superior to style-based predictions.

Regarding the differences between the $^{14}$C date ranges and Enoch's script style-based estimates, the mean absolute error (MAE) is 30.7 years. The MAE drops to 27.9 years when minor peaks are ignored (see illustration 26 in S8 Appendix). Minor peaks concern small secondary peaks in 12 cases, which are mostly smaller than 3.5% of maximum peak value except for two exceptions (5.2% in 4Q2 and 9.4% in 4Q416; see tables in S5 Appendix). In manuscript dating, MAE is commonly used [37] for evaluation of a regression method. The difference with rms error is limited [38]. With the chosen $2\sigma$ ($^{14}$C) and $1\sigma$ (AI) bandwidths,

the error for the leftmost margin is 6.4 years while for the rightmost margin it equals −38.4 years, indicating that Enoch's style-based estimate range ends earlier than the $^{14}$C range. For each sample, the date ranges of the two information sources have partial to full overlap with an average of 88.8%. For Ithaca [39], AI and epigraphy were similarly used as two heterogeneous information sources to predict dates for ancient Greek inscriptions. Their prediction provides an average distance of 29.3 years from the target dating brackets, with a median distance of 3 years based on the totality of texts. We also aim for date prediction, but, unlike Ithaca, we utilize three information sources: $^{14}$C, shape-based writing style analysis (AI), and palaeography, the latter only in the case of bimodal $^{14}$C evidence.

Qualitatively, Enoch's style-based predictions largely follow the $^{14}$C results, even though the validation samples (rows) are in no way present in the training data. In the range 300–50 BCE, Enoch's estimates provide an improved granularity compared to $^{14}$C. For samples 5/6Hev1b, Mas1k, and XHev/Se2, the style-based estimate is earlier and more uncertain. However, all of them overlap with $^{14}$C estimates. Interestingly, although 5/6Hev1b has the widest predicted range from Enoch, the palaeographic estimate still falls within the overlap region. In addition, 11Q5 shows that in the late date range, a fairly certain style-based date estimate above 100 CE can also be achieved. This may go against historical reconstructions according to which the scrolls were hidden in the Qumran caves before the summer of 68 CE [40]. Yet, we did not impose here a chronological limit on the model, because of the $^{14}$C result for 11Q5, and in order to examine the possibility of style continuation after 70 CE.

Fig 1 shows the general result that, on average, $^{14}$C date ranges and Enoch's predictions indicate older dates than traditional palaeographic estimates. Only 4Q201 and 11Q5 have older palaeographic date estimates, although there is an overlap with the $^{14}$C results (see Sect 4.1.1 in S4 Appendix).

## 4 Exploration of style-based dating on unseen manuscripts

The general recipe for Enoch's analysis of manuscript images is presented in Table 1. Before applying this to other previously undated scrolls, we first tried out a known mediaeval benchmark data set of charters, MPS [32], with success [41]. We then applied the trained Enoch model to a collection of 135 unseen manuscripts from the ca. 1000 Dead Sea Scrolls to explore the viability of style-based dating at this stage (S7 Appendix). Enoch, thus, produces an empirical evaluation which modifies a previously uniform date expectancy distribution to a curvilinear distribution with some dates becoming more likely, others becoming less likely for a sample. Like the OxCal program for $^{14}$C, Enoch delivers probabilities on dates as well as the corresponding error estimates. This is advanced in comparison to older, more primitive methods which only provide, e.g., a date point as an answer. In the analysis of our results both the (1) likelihood of a date point and the (2) estimation reliability of that point need to be taken into account. These are the first published results on date estimation for this collection of manuscripts. Future research, with more data and improved images, may be directed at further validation and refinement.

Table 2 summarizes the palaeographic post-hoc evaluation of Enoch's date predictions for 135 undated manuscripts. Expert palaeographers among the article's authors evaluated the style-based date predictions, condensing the prediction into two main categories: *realistic* and *unrealistic*, the latter subdivided into *too old* and *too young*. As can be seen in Table 2, 107 (79%) of the undated manuscripts were dated realistically and 28 unrealistic predictions (21%) were divided between too old (46%) and too young (39%), according to the palaeographers (S7 Appendix). With this sample size, the confidence margin is 7% at $\alpha = 0.05$. Enoch's date prediction task is not a 50/50, binary decision task but regressive, with many possible

**Table 1. Style-based date prediction recipe for Enoch.**

1. **Select and crop** the relevant manuscript images based on scholarly identification criteria;
2. In the images, perform a separation of the ink trace from the material background texture by using a deep-learning-based **U-net variant** for **multispectral image-intensity binarization** [26];
3. For each manuscript, compute **two shape descriptors**: a histogram of allographic fraglet occurrence and a histogram of angular co-occurrence along the ink-trace edges [29,30];
4. **Adjoin** the two **feature vectors**, properly weighted, to a single handwriting-style vector [43];
5. In order to decorrelate the features, avoid collinearity, and minimize the necessary number of parameters in the next stage, perform a strong **dimensionality reduction (PCA, 20 dimensions)**.
6. Take the **14C-dated manuscript-image samples** for **training** Enoch as a style-based **Bayesian ridge-regression model** with a scalar date estimate as the target output. In this training, augment the image data set by using random elastic morphing to obtain a sufficient and balanced number of examples per 14C-dated reference. This step is an essential, new contribution that allows a merger of 14C-based and style-based information in the date estimation. For validating Enoch, use the **leave-one-out** approach: each sample that is under evaluation does not occur in the training data;
7. **Predict**: style-based dates for **undated manuscripts**.

**Table 2. Expert evaluation of Enoch's date predictions**

| Prediction is: | Subcategory | Manuscript count | | Percentage |
|---|---|---|---|---|
| **Realistic** | - | - | 107 | 79.26% |
| **Unrealistic** | Indecisive | 4 | 28 | 20.74% |
| | Too old | 13 | | |
| | Too young | 11 | | |
| **Total manuscripts** | - | - | 135 | 100.00% |

years in the interval 300 BCE–200 CE. Assuming a coarseness of 25 years, as in the MPS project [32], the date range would consist of 20 bins, with a 5% prior-probability hit rate. A success rate of 79% is unlikely to be accidental. A binomial test achieves a p-value of $4.44e{-}12$ so that the observed result is highly unlikely to have occurred by chance alone.

Previously, we demonstrated that two scribes were at work in the Great Isaiah Scroll [6]. Now, Enoch shows that there is no temporal difference between the two halves of the manuscript as if one part were written significantly later than the other. On the contrary, both scribes are estimated to have worked on their respective part of the scroll of 1QIsa$^a$ in the same period. Fig 2 shows that Enoch dates the two halves consistently between 180–100 BCE.

## 5 The Enoch approach to dating ancient manuscripts

To our knowledge, Enoch is the first complete machine-learning-based model that employs raw image inputs to deliver probabilistic date predictions for handwritten manuscripts utilizing the probability distribution from 14C output, and that is completed by palaeographic input while ensuring transparency and interpretability through its explainable design. Also, Enoch's integration of multiple dating methods yields a strongly improved value of sources of evidence and allows for a mutual confirmation of evidence from the two sources—physical (material) and geometric (shape-based). As an illustrative example, samples 4Q259 and 4Q319 show that Enoch can accurately find a similar date estimate for the same writing style. The accepted $2\sigma$ calibrated range of 4Q259 was used to train Enoch. 4Q259 contains text that is part of the so-called Rule of the Community. 4Q319 contains a calendrical text. Due to perceived generic differences, 4Q319 received a separate classification number but is materially actually part of the same manuscript as 4Q259 [42]. Fig 3 shows that Enoch was able to give a date prediction estimate for 4Q319 that is similar to the accepted $2\sigma$ calibrated range of 4Q259 (see Sect 7.5 in S7 Appendix).

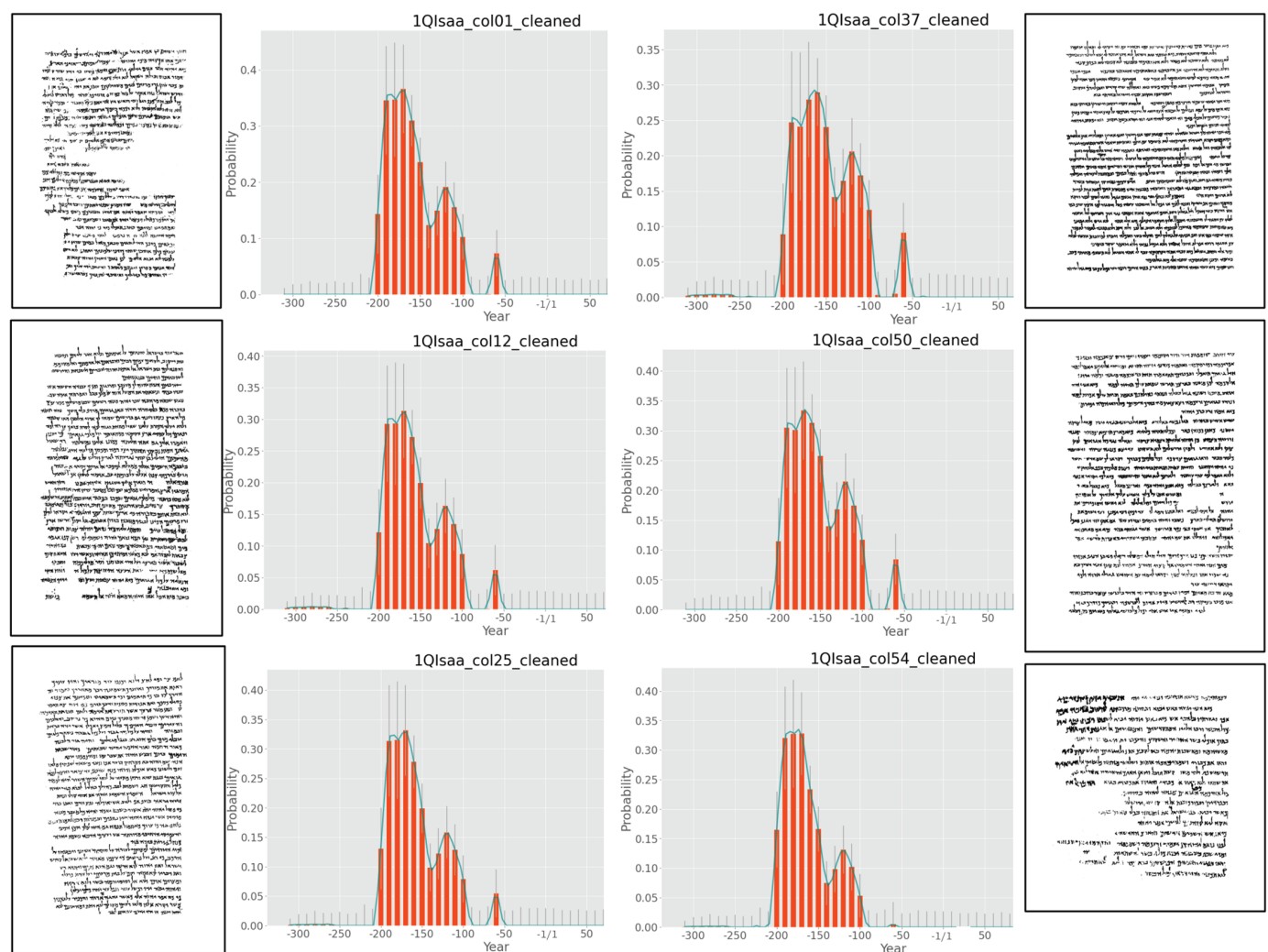

**Fig 2. Enoch's date prediction plots for 6 of the 54 columns from the two halves of 1QIsaᵃ.** The left 3 columns are from the first half of the manuscript, the right 3 columns are from the second half of the manuscript.

This study in style-based date prediction using the Enoch approach is a first step. The advantage of the Enoch model is that it provides quantified objectivity to palaeography, reducing the method's subjectivity and the role of tacit expert knowledge, by offering a limited number of probability-based options on empirical grounds, both physical ($^{14}$C) as well as geometric (shape-based analysis) evidence, that can assist palaeographers to substantiate, sharpen, or modify their own estimate for an individual manuscript. Also, the methods underpinning Enoch can be used for date prediction in other partially-dated manuscript collections. Finally, we did not take any model that is already available, but we developed a robust model that can (1) predict dates using only a very small amount of data, i.e., 24 sample or data points, (2) deal with uncertainty, and (3) provide explainability.

It could be argued that the style-based predictions are influenced by the $^{14}$C-based training of the model. However, the leave-one-out validation results indicate that unseen samples obtain their interpolated position on the time axis based on the detected handwriting style in

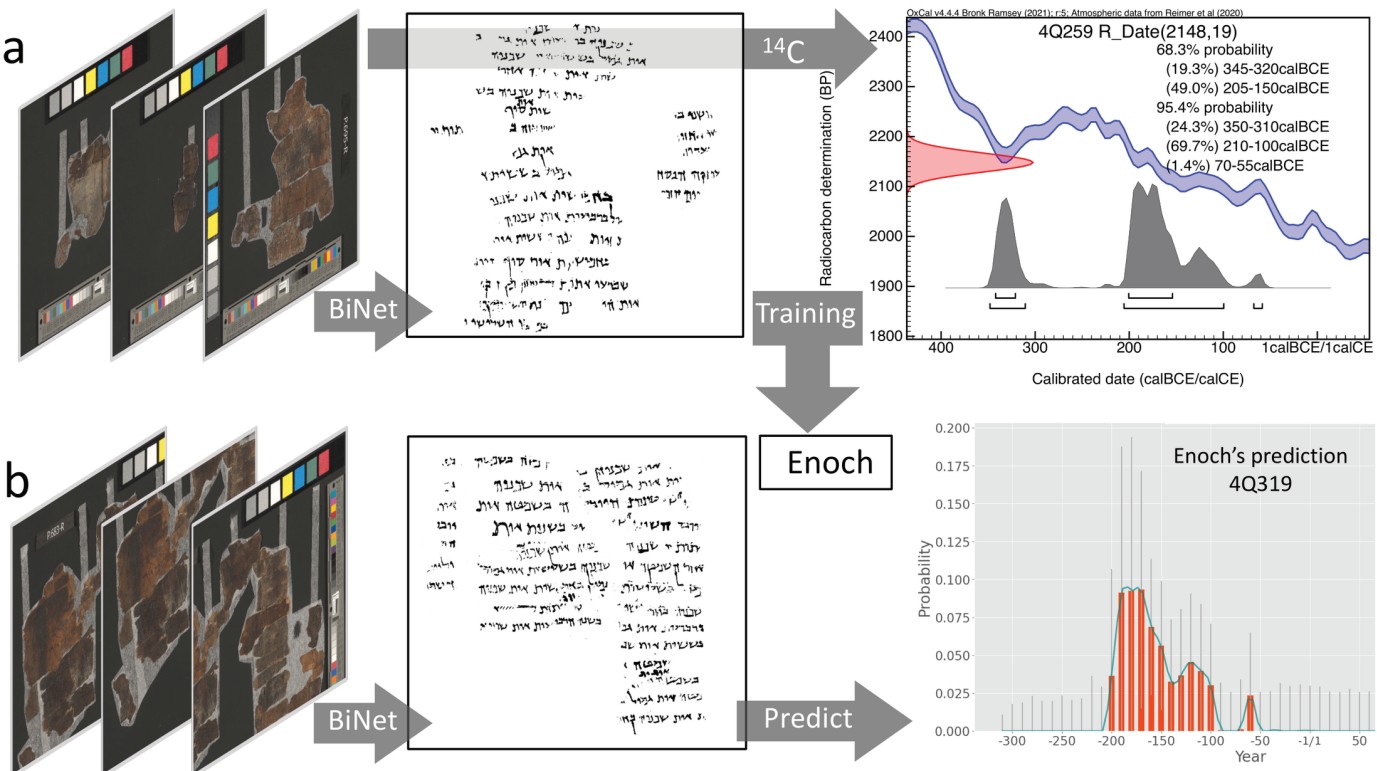

**Fig 3. Enoch's date prediction estimate for 4Q319.** *(a)* from full spectrum colour image to binarized image to ¹⁴C plot for 4Q259 that went into the training of Enoch. *(b)*, from full spectrum colour image to binarized image to Enoch's date prediction plot for 4Q319 (see also illustration 9 in S5 Appendix). Red bars represent the probability of each date bin. The blue curve shows the smoothed distribution. Grey spikes indicate the local uncertainty of the estimate.

the images. The placement of an unseen sample on the time axis is not fundamentally constrained. Any date in the time range of 300 BCE–200 CE could have been reached, looking at all style-based dates empirically covered by the model.

In this study, we have avoided using palaeographic estimates as target values for machine learning because our goal is to provide physical (material) and geometric (shape-based) evidences for manuscript dating. While the use of palaeographic estimates as target values for machine learning is technically possible, we consider it too risky, given the existing uncertainties and lack of consensus associated with the precise dating of individual manuscripts.

In its manuscript analysis, Enoch differs from traditional palaeographic approaches. Enoch emphasizes shared characteristics and similarity matching between trained and test manuscripts, whereas traditional palaeography focuses on subtle differences that are assumed to be indicative for style development. Combining dissimilarity matching and adaptive reinforcement learning can uncover hidden patterns. This interdisciplinary fusion may enrich our understanding of textual content, material properties, and historical context, leading to enhanced interpretations of the past. This remains a task for the future. New ¹⁴C evidence or, with new discoveries, a whole range of date-bearing manuscripts can be added to Enoch's training data for further refinement and precision, continuously improving accuracy. The consequences of each newly added manuscript sample to the Dead Sea Scrolls ¹⁴C reference collection can now easily be computed using the Enoch approach.

Although the limited data were insufficient for a full deployment of deep-learning in the prediction task (S6 Appendix), future research needs to address the problems of sparse labeling and high dimensionality. It is to be expected that new solutions will appear here, because these problems are encountered in many application domains. If palaeographers are willing to accept the use of 'black box', pre-trained deep-learning models that are based on completely extraneous large image and photograph collections, future research may be directed at adapting the output of such models to the vectorial regression-based date-prediction task that is proposed in the current article.

## 6 Aramaic/Hebrew script development in ancient Judaea

The results of this study lead to four novel insights into Aramaic/Hebrew script development during the period under consideration and the date of individual manuscripts.

First, [14]C date ranges and Enoch's style-based estimates are overall older than previous palaeographic estimates. These older dates for the scrolls are realistic. Hasmonaean-type manuscripts have accepted $2\sigma$ calibrated ranges that allow for older dates in the first half of the second century BCE, and sometimes slightly earlier, instead of only circa 150–50 BCE. There are no compelling palaeographic or historical reasons that preclude these older dates as reliable time markers for the 'Hasmonaean' script. This also applies to the accepted $2\sigma$ calibrated range for 4Q70 and its 'Archaic' script.

Second, 'Herodian' script emerged earlier than previously thought. This suggests that the 'Hasmonaean' and 'Herodian' scripts were not transitioning from the mid-first century BCE onward, but that they existed next to each other at a considerably earlier date.

Third, this novel approach of palaeography leads to a new chronology of the scrolls that impacts our understanding of the history of ancient Judaea and the people behind the scrolls. Hypotheses about whether the movement behind the scrolls originated in the second or first century BCE will need to be reconsidered in light of Enoch's second-century BCE date predictions for Hasmonaean-type manuscripts such as 1QS and 4Q163 (S7 Appendix), bearing texts that are regarded typical for the movement. Scholars often assume that the rise and expansion of the Hasmonaean kingdom from the mid-second century BCE onward caused a rise in literacy and gave a push to scribal and intellectual culture. Yet, the results of this study attest to the copying of multiple literary manuscripts before this period. One example is 4Q109, a copy of the biblical book of Ecclesiastes, a book which scholars tentatively date to the end of the third century BCE [15], for which Enoch gives a third-century BCE date prediction (S7 Appendix), close to Archaic-type manuscripts such as 4Q52 and 4Q70—copies of the biblical books of Samuel and Jeremiah.

Fourth, this study's [14]C result for 4Q114 and Enoch's date prediction for 4Q109 now establish these to be the first known fragments of a biblical book from the time of their presumed authors [15].

The results of this study thus dismantle unsubstantiated historical suppositions and chronological limitations, and call into question the validity of the default model's relative typology. This relative typology can only be maintained with restrictions. The spread of the Hasmonaean-type manuscripts over the timeline does not affect the default relative typology in a major way, but the older, second-century BCE date ranges of the Herodian-type manuscripts challenge the relative typology. More research is needed to solve this issue.

## Supporting information

**S1 Appendix. The dating problem of the Dead Sea Scrolls.**
(PDF)

**S2 Appendix. Radiocarbon dating of the Dead Sea Scrolls.**
(PDF)

**S3 Appendix. $^{14}$C determinations and calibrated date plots.**
(PDF)

**S4 Appendix. Palaeography and radiocarbon dating of the Dead Sea Scrolls.**
(PDF)

**S5 Appendix. Artificial intelligence (AI) in dating the scrolls.**
(PDF)

**S6 Appendix. On the use of pre-trained deep learning methods for image-based dating.**
(PDF)

**S7 Appendix. Enoch's date predictions for 135 previously undated manuscripts.**
(PDF)

**S8 Appendix. Comparative plots for different information sources.**
(PDF)

**S9 Appendix. List of images for different tests.**
(PDF)

**S10 Appendix. Radiocarbon sample information.**
(PDF)

**S11 Appendix. Data-sheet radiocarbon runs.**
(PDF)

**S12 Appendix. Worksheet of comparative data for 2$\sigma$ $^{14}$C dates and traditional palaeographic estimates.**
(PDF)

## Acknowledgments

The authors thank P. Shor, J. Uziel, T. Bitler, H. Libman, B. Riestra, O. Rosengarten, and S. Halevi at the Dead Sea Scrolls Unit of the Israel Antiquities Authority (IAA) and E. Boaretto (advisor to the IAA from the Weizmann Institute of Science, Jerusalem) for providing physical samples and multispectral images of the scrolls—courtesy of the Leon Levy Dead Sea Scrolls Digital Library; Brill Publishers for the Dead Sea Scrolls images from the Brill Collection; A. Aerts-Bijma and D. Paul for handling and measuring the $^{14}$C samples at the Center for Isotope Research (Groningen); S. Legnaioli for the Raman analyses performed at the CNR-ICCOM (Pisa); A. Krauss and T. van der Werff for their contributions to developing and testing Enoch; L. Bouma for cleaning images; D. Longacre, G. Hayes, A.W. Aksu, H. van der Schoor, C. van der Veer, and M. van Dijk for their contributions to preparing images for training Enoch; M.W. Dee for advising on and inspecting the code and data acquisition process from OxCal to the Enoch model at the Center for Isotope Research (Groningen). All necessary permits were obtained for the described study, which complied with all relevant regulations.

## Author contributions

**Conceptualization:** Mladen Popović, Maruf A. Dhali, Lambert Schomaker.

**Data curation:** Mladen Popović, Maruf A. Dhali, Lambert Schomaker, Johannes van der Plicht, Kaare Lund Rasmussen, Jacopo La Nasa, Ilaria Degano, Maria Perla Colombini.

**Formal analysis:** Maruf A. Dhali, Lambert Schomaker, Johannes van der Plicht, Kaare Lund Rasmussen, Jacopo La Nasa, Ilaria Degano, Maria Perla Colombini.

**Funding acquisition:** Mladen Popović.

**Investigation:** Mladen Popović, Maruf A. Dhali, Lambert Schomaker, Johannes van der Plicht, Kaare Lund Rasmussen, Jacopo La Nasa, Ilaria Degano, Maria Perla Colombini, Eibert Tigchelaar.

**Methodology:** Mladen Popović, Maruf A. Dhali, Lambert Schomaker, Johannes van der Plicht, Kaare Lund Rasmussen.

**Project administration:** Mladen Popović.

**Resources:** Mladen Popović, Lambert Schomaker, Johannes van der Plicht, Kaare Lund Rasmussen, Maria Perla Colombini.

**Software:** Maruf A. Dhali, Lambert Schomaker.

**Supervision:** Mladen Popović, Lambert Schomaker, Johannes van der Plicht, Maria Perla Colombini.

**Validation:** Mladen Popović, Maruf A. Dhali, Lambert Schomaker, Johannes van der Plicht, Kaare Lund Rasmussen, Jacopo La Nasa, Ilaria Degano, Maria Perla Colombini.

**Visualization:** Mladen Popović, Maruf A. Dhali, Lambert Schomaker, Johannes van der Plicht, Kaare Lund Rasmussen, Jacopo La Nasa, Ilaria Degano, Maria Perla Colombini.

**Writing – original draft:** Mladen Popović, Maruf A. Dhali, Lambert Schomaker, Johannes van der Plicht, Kaare Lund Rasmussen, Jacopo La Nasa, Ilaria Degano, Maria Perla Colombini, Eibert Tigchelaar.

**Writing – review & editing:** Mladen Popović, Maruf A. Dhali, Lambert Schomaker, Johannes van der Plicht, Kaare Lund Rasmussen, Jacopo La Nasa, Ilaria Degano, Maria Perla Colombini, Eibert Tigchelaar.

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
