## [Decision Letter · Decision Letter 0]

23 Jan 2025

PONE-D-24-53510Dating ancient manuscripts using radiocarbon and AI-based writing style analysisPLOS ONE

Dear Dr. Popović,

Thank you for submitting your manuscript to PLOS ONE. After careful consideration, we feel that it has merit but does not fully meet PLOS ONE’s publication criteria as it currently stands. Therefore, we invite you to submit a revised version of the manuscript that addresses the points raised during the review process.

**Editor's Response:**

Reviewers appreciate the potential of your innovative approach to dating historical scrolls using AI and radiocarbon data. However, they highlight several areas for improvement before publication. Key concerns include the model's reliance on its training data's chronological range and the unclear integration of palaeography. Reviewers recommend clarifying these points, enhancing reproducibility (e.g., specifying Python versions and providing virtual environment instructions), and addressing minor issues such as missing figures and redundant text.

We find the study scientifically valuable and encourage you to address these revisions to strengthen the manuscript for publication.

We look forward to receiving your revised manuscript.

Kind regards,

Carlos P. Odriozola, Ph.D

Academic Editor

PLOS ONE

Journal Requirements:

2. In your manuscript, please provide additional information regarding the specimens used in your study. Ensure that you have reported human remain specimen numbers and complete repository information, including museum name and geographic location.

For more information on PLOS ONE's requirements for paleontology and archeology research, see https://journals.plos.org/plosone/s/submission-guidelines#loc-paleontology-and-archaeology-research.

“This project has received funding by the European Research Council under the European Union’s Horizon 2020 research and innovation programme under grant agreement no. 640497 (HandsandBible). M.Popović and E.Tigchelaar were also supported by NWO, Netherlands Organisation for Scientific Research, and FWO, the Research Foundation - Flanders (SV-15-29).”

“This project has received funding by the European Research Council under the European Union’s Horizon 2020 research and innovation programme under grant agreement no. 640497 (HandsandBible). M.P. and E.T. were also supported by NWO, Netherlands Organisation for Scientific Research, and FWO, the Research Foundation - Flanders (SV-15-29).”

“This project has received funding by the European Research Council under the European Union’s Horizon 2020 research and innovation programme under grant agreement no. 640497 (HandsandBible). M.Popović and E.Tigchelaar were also supported by NWO, Netherlands Organisation for Scientific Research, and FWO, the Research Foundation - Flanders (SV-15-29).”

6. Please upload a copy of Figure 12, 25 and 29, to which you refer in your text on page 4 and 5. If the figure is no longer to be included as part of the submission please remove all reference to it within the text.

7. Please include a copy of Table 19, 20, 21 and 10 which you refer to in your text on page 3 and 5.

8. Please review your reference list to ensure that it is complete and correct. If you have cited papers that have been retracted, please include the rationale for doing so in the manuscript text or remove these references and replace them with relevant current references. Any changes to the reference list should be mentioned in the rebuttal letter that accompanies your revised manuscript. If you need to cite a retracted article, indicate the article’s retracted status in the References list and also include a citation and full reference for the retraction notice.

Reviewers' comments:

Reviewer's Responses to Questions

**Comments to the Author**

1. Is the manuscript technically sound, and do the data support the conclusions?

Reviewer #1: Yes

Reviewer #2: Yes

2. Has the statistical analysis been performed appropriately and rigorously? 

Reviewer #1: Yes

Reviewer #2: Yes

3. Have the authors made all data underlying the findings in their manuscript fully available?

Reviewer #1: Yes

Reviewer #2: Yes

4. Is the manuscript presented in an intelligible fashion and written in standard English?

Reviewer #1: Yes

Reviewer #2: Yes

5. Review Comments to the Author

Reviewer #1: The authors' aim has been to use the geometric features derived from the writing style as vectors to construct a regression function that points to a probabilistic range that allows dating scrolls in a specific chronological range.

The training dataset has been constructed from the dating of 30 manuscripts from which 24 valid C14 dates have been obtained for training. The images of the scrolls were binarised using a variant of U-Net and used as input for the model. Of the 75 processed images, 62 were used for training and 13 for evaluation using a leave-one-out strategy, appropriate given the size of the data.

The idea of using features based on the shape of handwriting to train a supervised regression model oriented to a chronological range based on c14 dates is indeed a clever way of looking for patterns in the data using novel computational methods that has the potential to bring a greater objectivity and a higher degree of resolution to the study of historical questions in the specific period and subject under consideration and can therefore be of help to palaeographers and historians in their decision-making and contribute to historical debates.

The model is fully reproducible, resulting in a folder with tabular and graphical outputs that match those presented in the text.

Particularly the contribution of new empirical data (c14 dates) is quite relevant to the problem of accurately reconstructing historical developments from the 4th to the 2nd century BCE and fill the chronological gap between the third century BCE and the second century CE (according to the authors) .

I am not an expert on the particular subject on which the ML techniques have been applied, so my comments are limited to a critical assessment of the methods used on an archaeological problem given my experience in that field.

I have some comments which I will describe below and which lead me to suggest some minor revisions before publication:

My main concern is how much weight the model actually contributes to the conclusions drawn. Let me explain: If the model has been trained with c14 dates in a specific chronological range. Therefore, when making out-of-sample estimates, the model will only be able to give results in the chronological ranges within which it has been trained. Thus, it is possible that some of the undoubtedly interesting conclusions, such as the greater antiquity of some materials (and their historical consequences) are strictly due to the empirical data provided and not to the results of the AI approach. With a larger sample set for training, perhaps these conclusions will change?

This leads me to suggest a consideration of nuancing some of the conclusions drawn in the paper as these observed correlations between chronological ranges and features derived from writing style, while undoubtedly relevant and contributing to reducing uncertainty about the particular problem, have limitations that could be made more explicit and should serve to prevent causal conclusions.

- Line(75) It could be dispensed with as it does not add much to the debate.

- (Line 167) I cant find fig 29.

- (Line 1529-1532 S7). Did the authors of the article perform the post-hoc evaluation? If so, more clarity is needed in the main text on how the assessment has been carried out, as the issue has methodological implications as it is the baseline from which the final reliability of the model is assessed when dealing with out-of-sample data and may be susceptible to confirmation bias.

- (Line 234 Main text) What are the multiple dating methods integrated in the model? As I understand it, the shape-based features derived from the writing style (geometric evidence) have been extracted, binarized and used as input to construct a supervised dataset whose response variable is a date interval (2σ) according to the set of radiocarbon dates (phyisical evidence) presented, and that the results have been validated against the default palaeographic method. It is not clear how other dating methods(palaeography) are integrated into the model. My concern is compounded by the statement in line (258) about avoiding palaeographic estimates as target values and the statement in line (890 S4.2) that says that ‘In this study, we combine palaeography and radiocarbon dating to train our date prediction model’. Just after, in line (924 S 4.2) it is confirmed that ‘To train our artificial intelligence-based date prediction model, we used the accepted 2σ calibrated data from 24 of the 26 valid 14 C results". In Appendix S 4.2, although the problem associated with the qualitative nature of the palaeographic method for estimating the age of the scrolls is explained, it is not made clear how this method has been integrated into the training of the model. Therefore, it seems to me necessary to better explain how palaeography has been used in the model's predictor space, if at all, or to disambiguate the different references to the dating methods used within the model, as well as to rephrase the title of S4.2.

Technical comments:

Although the reproduction of the procedures presented requires a certain level of expertise for which the resources are sufficiently well presented, possible non-technically skilled stakeholders may benefit from having a static version that presents the results alongside the code.

The inclusion of indications for the creation of a virtual environment as well as the specific version of python whitin the README file would benefit the reproducibility of the exercise.

The work therefore presents in a general way novel knowledge that can contribute to advance the knowledge of the specific subject and is of general scientific and historical interest.

Reviewer #2: This manuscript presents a novel use of AI based on radiocarbon dating and handwriting style to determine or correct the dates estimated by the palaeographic studies of the so-called Dead Sea Scrolls. It introduces Enoch, an AI-based prediction model that the authors have trained and tested with these manuscripts, demonstrating its potential, which this reviewer understands to be of significant interest (even to be trained and used in other archaeological/historical contexts where dating based on writing analysis can be corrected).

The study presents results of an original research. With regard to materials and methods, the authors are thorough in describing the manuscript selection processes, sampling, laboratory analysis, and data treatment. Radiocarbon analysis protocols have been adapted to solve the difficulties on dating the fragile samples avoiding contamination.

The main difficulty in dating the scrolls lies in the uncertainty inherent in the two methods used, the discrepancies on writing style analysis and the limitations of the radiocarbon dating method and its calibration. The authors dedicate several pages in the supplementary material (S1) to demonstrating the contradictions of palaeography-based dating and the necessity of tools like Enoch.

The authors state that radiocarbon dates are “more reliable time markers” and “palaeographic estimates do not provide absolute or fixed dates” (lines 572-576, Appendix S4.1). In addition, they “compare the radiocarbon dates with previous palaeographic estimates only in a general sense, not as a rigid application of these estimates” (lines 520-521, Appendix S4.1). However, when working with calibrated radiocarbon dates, the selection or rejection of the C14 temporal ranges used to train Enoch are based on very specific palaeographic temporal proposals (e.g., Appendix S4.1.1). In this regard, the dependence of the C14 dating use from palaeography might appear as an important contradiction.

Concerning this, author’s arguments are better explained in Appendix S5.The statistical calculations and the functioning of Enoch are discussed in detail by the authors. The limitations of the sample size (24 radiocarbon dated scrolls) is explained too. Conclusions are supported by the data.

Finally, in addition to the 95 pages of supplementary material, the authors provide access to other data, such as the OxCal codes, raw data or a video where the rationale behind the research and the control tests conducted can be better understood.

In the opinion of this reviewer, the article should be published with only a couple of minor improvements:

1- In the case of 5/6 Hev1b, the discrepancies between the C14 results and the paleographic estimates are explained (lines 711-716, S4.1.1). However, the discrepancy between those dates and the ones proposed by Enoch, which is clearly shown in Figure 1, is not sufficiently detailed.

2- For practical reasons, table 21 should include Q-numbers.

6. PLOS authors have the option to publish the peer review history of their article (what does this mean?). If published, this will include your full peer review and any attached files.

Reviewer #1: No

Reviewer #2: No

---

## [Author Response · Author response to Decision Letter 1]

4 Mar 2025

Our response is in the rebuttal letter

---

## [Decision Letter · Decision Letter 1]

4 Apr 2025

Dating ancient manuscripts using radiocarbon and AI-based writing style analysis

PONE-D-24-53510R1

Dear Dr. Popović,

We’re pleased to inform you that your manuscript has been judged scientifically suitable for publication and will be formally accepted for publication once it meets all outstanding technical requirements.

Kind regards,

Carlos P. Odriozola, Ph.D

Academic Editor

PLOS ONE

Additional Editor Comments (optional):

Reviewers' comments:

Reviewer's Responses to Questions

**Comments to the Author**

1. If the authors have adequately addressed your comments raised in a previous round of review and you feel that this manuscript is now acceptable for publication, you may indicate that here to bypass the “Comments to the Author” section, enter your conflict of interest statement in the “Confidential to Editor” section, and submit your "Accept" recommendation.

Reviewer #1: All comments have been addressed

Reviewer #2: All comments have been addressed

2. Is the manuscript technically sound, and do the data support the conclusions?

Reviewer #1: Yes

Reviewer #2: (No Response)

3. Has the statistical analysis been performed appropriately and rigorously? 

Reviewer #1: Yes

Reviewer #2: (No Response)

4. Have the authors made all data underlying the findings in their manuscript fully available?

Reviewer #1: (No Response)

Reviewer #2: (No Response)

5. Is the manuscript presented in an intelligible fashion and written in standard English?

Reviewer #1: Yes

Reviewer #2: (No Response)

6. Review Comments to the Author

Reviewer #1: The comments have been addressed by the authors and some minor modifications have improved the interpretability of the work.

Reviewer #2: (No Response)

7. PLOS authors have the option to publish the peer review history of their article (what does this mean?). If published, this will include your full peer review and any attached files.

Reviewer #1: No

Reviewer #2: No

---

## [Editor Report · Acceptance letter]

PONE-D-24-53510R1

PLOS ONE

Dear Dr. Popović,

I'm pleased to inform you that your manuscript has been deemed suitable for publication in PLOS ONE. Congratulations! Your manuscript is now being handed over to our production team.

Kind regards,

on behalf of

Dr. Carlos P. Odriozola

Academic Editor

PLOS ONE